# Molecular Engineering of Virus Tropism

**DOI:** 10.3390/ijms252011094

**Published:** 2024-10-15

**Authors:** Bo He, Belinda Wilson, Shih-Heng Chen, Kedar Sharma, Erica Scappini, Molly Cook, Robert Petrovich, Negin P. Martin

**Affiliations:** 1Viral Vector Core, Neurobiology Laboratory, National Institute of Environmental Health Sciences, Research Triangle Park, NC 27709, USA; bo.he@nih.gov (B.H.); wilson3@niehs.nih.gov (B.W.); chens3@niehs.nih.gov (S.-H.C.); 2Genome Integrity and Structural Biology Laboratory, National Institute of Environmental Health Sciences, Research Triangle Park, NC 27709, USA; kedar.sharma@nih.gov (K.S.); cook1@niehs.nih.gov (M.C.); petrovi1@niehs.nih.gov (R.P.); 3Fluorescent Microscopy and Imaging Center, Molecular and Cellular Biology Laboratory, National Institute of Environmental Health Sciences, Research Triangle Park, NC 27709, USA; scappinie@niehs.nih.gov

**Keywords:** viral vectors, viral application, pseudotyping virus, virus envelope, virus envelope chimera, AAV serotype, AAV variant, capsid, AAV, lentivirus, retrovirus, g-deleted rabies virus, HSV

## Abstract

Engineered viral vectors designed to deliver genetic material to specific targets offer significant potential for disease treatment, safer vaccine development, and the creation of novel biochemical research tools. Viral tropism, the specificity of a virus for infecting a particular host, is often modified in recombinant viruses to achieve precise delivery, minimize off-target effects, enhance transduction efficiency, and improve safety. Key factors influencing tropism include surface protein interactions between the virus and host-cell, the availability of host-cell machinery for viral replication, and the host immune response. This review explores current strategies for modifying the tropism of recombinant viruses by altering their surface proteins. We provide an overview of recent advancements in targeting non-enveloped viruses (adenovirus and adeno-associated virus) and enveloped viruses (retro/lentivirus, Rabies, Vesicular Stomatitis Virus, and Herpesvirus) to specific cell types. Additionally, we discuss approaches, such as rational design, directed evolution, and in silico and machine learning-based methods, for generating novel AAV variants with the desired tropism and the use of chimeric envelope proteins for pseudotyping enveloped viruses. Finally, we highlight the applications of these advancements and discuss the challenges and future directions in engineering viral tropism.

## 1. Introduction

Viral tropism refers to the specific preference of a virus to infect and replicate within certain cell types, tissues, or host organisms. This selective infection is governed by several factors, including the presence of specific receptors on the host-cell surface, the compatibility of the intracellular environment for viral replication, and the virus’s ability to evade the host’s immune system [1]. When a virus fails to bind to the surface of the host-cell or the host lacks the necessary machinery required for virus replication, protein synthesis, and virion assembly, the infection fails. Viral tropism can also be defined by the target species, with some viruses infecting a narrow range of species, while others have a broad host range. Zoonotic viruses, such as the Influenza virus, can cross species barriers, infecting multiple species, including humans, which significantly contributes to their pandemic potential [2]. Moreover, the ability of a virus to evade the host’s immune response is crucial for a successful infection and further propagation [3].

The interaction between viral and host-cell surface proteins (or molecules) is a key determinant of viral tropism. Successful attachment to a host-cell, facilitated by these interactions, is the initial step in the infection process. Examples of viral tropism include HIV, a lentivirus which targets CD4+ T cells, macrophages, and dendritic cells by binding to the CD4 receptor and co-receptors CCR5 or CXCR4 [4]; Influenza virus, which primarily infects respiratory epithelial cells by attaching to sialic acid receptors [5]; and Hepatitis B virus, which targets hepatocytes due to the specific receptors and favorable environment they provide for viral replication [6]. The Rabies virus, with its strong tropism for neurons, is capable of retrograde transport through the nervous system to the brain [7].

Understanding viral tropism is essential in defining disease pathogenesis, determining which tissues or organs are affected, and influencing the symptoms and severity of the disease. Tropism also plays a crucial role in how a virus spreads within a host and between hosts. For instance, a virus with a respiratory tract tropism is likely to spread through airborne transmission. By understanding a virus’s tropism, targeted therapies and vaccines can be developed, such as blocking the receptors a virus uses to enter cells to prevent infection [8].

Advances in virology have enabled researchers to engineer viral tropism, expanding the range of viral hosts or restricting them to specific cell types. Altering viral surface proteins or pseudotyping enveloped viruses are common strategies to modify tropism in non-enveloped and enveloped viruses, respectively. For instance, conjugating antibodies or peptides to viral surface proteins can direct viruses to target specific cancer cells. These engineered viruses are invaluable in therapeutic applications such as gene therapy, cancer treatment, and vaccine development, allowing for safer and more precise interventions [9]. In another case, baculovirus, an enveloped insect virus, was modified through pseudotyping with the Vesicular Stomatitis Indiana virus glycoprotein to BacMam (Thermo Fisher Scientific, Waltham, MA, USA) to deliver genes to mammalian brains (Figure 1). The engineered baculovirus delivers genes robustly, and the delivered genes express within hours of infection. BacMam allows for gene delivery to ex vivo brain cultures and the tissues of species that are short-lived in culture for biomedical research. Additionally, viral vectors are widely used as research tools to genetically alter cells for creating in vitro and in vivo models for studying gene function and diseases [10].

In this review, we provide an overview of strategies employed to alter the tropism of recombinant non-enveloped viruses (such as adeno-associated viruses and adenoviruses) and enveloped viruses (including retro/lentivirus, Rabies, the Vesicular Stomatitis Virus, and Herpesvirus). We will explore the latest advancements and considerations in designing capsid proteins and pseudotyping, which are critical strategies for creating powerful tools for cancer treatment, gene therapy, vaccine development, and generating biomedical tools.

## 2. Viral Envelopes and Capsids

Viruses can be divided into the two main categories of enveloped and non-enveloped based on the presence or absence of a lipid bilayer membrane on their outer shells. Enveloped and non-enveloped viruses differ in their structures, mode of infection, and stability. Both categories of viruses deliver their genome in a capsid shell. Capsids are typically composed of repeating protein subunits called capsomeres [12]. Enveloped viruses have an outer lipid bilayer membrane (envelope) that surrounds the viral capsid. This envelope is derived from the host-cell’s membrane (plasma membrane or internal membranes such as the Golgi apparatus or endoplasmic reticulum) during the budding process [13]. The envelope is studded with viral glycoproteins that facilitate attachment and entry of the virus into the host-cells by binding to specific surface receptors. Non-enveloped viruses lack this lipid envelope and instead rely on their capsid proteins to provide protection and help in the attachment to host-cells. The lipid envelope makes enveloped viruses more sensitive to environmental factors such as heat, desiccation, detergents, and solvents [13]. Most enveloped viruses are not airborne or have a short-lived lifespan in droplets and aerosols as compared to non-enveloped viruses. Airborne transmission of SARS-CoV-2 in exhaled respiratory aerosols and cough droplets is an example of an airborne enveloped virus [14]. Non-enveloped viruses are generally more resistant to environmental stressors, including varying temperatures, pH changes, and even some detergents [15]. Their robust capsid provides protection in various conditions. Adeno-associated viruses are an example of a non-enveloped virus that can withstand pH ranges of 3 to 9 in a range of temperatures based on their serotype [16]. This stability allows non-enveloped viruses to persist on surfaces for longer periods and be transmitted via routes such as the fecal–oral route, fomites, or contaminated water. Due to their fragility, enveloped viruses are typically transmitted through close contact, bodily fluids, or aerosols, where they are protected from harsh environmental conditions. Examples of enveloped viruses include the Influenza virus, HIV, Herpesviruses, coronaviruses (e.g., SARS-CoV-2), and the Hepatitis B virus. Non-enveloped viruses include adenoviruses, Poliovirus, Rhinoviruses, the Hepatitis A virus, and Noroviruses. Some enveloped viruses, like HIV and Influenza, undergo rapid antigenic variation in their surface glycoproteins, making it difficult for the immune system to mount an effective long-term response [17]. Non-enveloped viruses generally exhibit more stable antigenic properties compared to enveloped viruses, which means vaccines against non-enveloped viruses may offer longer-lasting protection.

To engineer the viral tropism of recombinant non-enveloped viruses, such as AAVs, researchers have developed innovative strategies such as rational design, directed evolution, and in silico and machine learning-based methods. These strategies and the use of chimeric envelope proteins for pseudotyping enveloped viruses will be discussed in the following sections.

## 3. Engineered Tropism of Recombinant Non-Enveloped Viruses (AAVs and Adenoviruses)

Non-enveloped viruses, particularly adenoviruses and AAVs, are valuable tools in gene delivery, offering flexibility and efficiency for research and therapeutic applications. Their ability to be engineered for specific targets makes them suitable for a range of gene delivery strategies. Non-enveloped viruses can efficiently deliver genetic material to a wide range of cells. AAVs, in particular, offer the potential for long-term expression of therapeutic genes with minimal immune responses. However, they have a limited genetic cargo capacity [18]. On the other hand, adenoviral vectors are capable of carrying large genetic cargos but can induce a strong immune response, which may limit their use in some gene therapy applications or require immunosuppressive strategies [19].

### 3.1. Adeno-Associated Viruses

Adeno-associated viruses (AAVs) are non-enveloped viruses that belong to the parvovirus family. They have a linear, single-stranded DNA (ssDNA) with a genome size of around 4.7 kilobases (kb). Due to their low pathogenicity, replication-defective, non-integrating expression profiles, recombinant AAV vectors have been actively pursued for tropism engineering. Today, AAVs are one of the most promising gene therapy vehicles for human diseases [20]. This was evidenced by the approvement of the first AAV gene therapy product, Alipogene Tiparvovec (Glybera), in Europe in 2012 [21], followed by Voretigene Neparvovec (Luxturna) in the United States in 2017 [22].

An AAV is a protein shell encapsulating a small single-stranded DNA genome which contains *Rep, Cap*, and *Aap* genes. The *Rep* gene encodes Rep78, Rep68, Rep52, and Rep40 proteins, which are required for viral genome replication and packaging. The *Cap* gene encodes the structural capsid proteins VP1, VP2 and VP3, which assemble into an icosahedral shape with a molar ratio of 1:1:10 (VP1:VP2:VP3). They not only protect the viral genome inside but are also involved in the molecular interaction between the ligands on the capsid and the receptors on the target cell membrane [23]. This interaction determines AAV tropism, the AAV’s ability to infect specific cell types or tissues. Binding to the host-cell surface receptors is the first step for many AAV infections. Different AAV serotypes, which vary with distinct capsid structures, have evolved to bind to specific cellular receptors, which vary across different tissues. In addition to the primary receptors, co-receptors and host factors also play important roles in facilitating viral entry and the infection of target cells. While a specific AAV serotype may use multiple receptors depending on the targeting tissues or cells, AAV tropisms in vivo are also dictated by its administration routes and the interaction with serum proteins [24,25]. For example, Heparan sulfate proteoglycan (HSPG) was identified as a primary receptor for the binding of AAV2, AAV3 and AAV6 to the target cell membrane [26]. However, AAV2 also needs cellular co-receptors, such as human fibroblast growth factor receptor 1 (FGFR1) and αV-β5 Integrin [27,28]. Several AAV3 co-receptors were found, including FGFR1 and hepatocyte growth factor receptor (HGFR) [29], while AAV6 uses epidermal growth factor receptor (EGFR) as a co-receptor to gain entry into the host-cell [30]. Both AAV8 and AAV9 use Laminin as the primary receptor and Laminin receptor (LamR) as a co-receptor, respectively [31]. AAV9 can also use the terminal N-linked galactose as a primary receptor [32]. An adeno-associated virus receptor (AAVR) was identified as an universal receptor for multiple AAV serotypes [33].

AAV capsid sequences and the genomic DNA cargo have been placed under intense investigation. AAV serotypes refer to the different capsid proteins of the naturally occurring AAV virus, such as AAV1-AAV9. Different serotypes have different preferences for infecting specific types of cells in different tissues. AAV variants, however, are genetically modified versions of naturally occurring serotypes. Today, recombinant AAVs (rAAVs) are the leading platform for in vivo gene delivery and clinical gene therapies. Currently, there are over 255 AAV-mediated gene therapy clinical trials. Seven AAV-based gene therapy products have received regulatory approval [34]. Nevertheless, despite these achievements, there are many concerns in preclinical and clinical studies, such as delivery efficiency, packaging optimization, and immune response when a high-dose viral therapy is utilized. Therefore, an exploration of novel AAV variants with improved properties is urgently needed. In recent years, there have been numerous successful attempts to engineer AAV capsids that have thus altered specific AAV properties, such as cellular tropism, transduction efficiency, and immunogenicity. The current methods for capsid engineering could be categorized into rational design, directed evolution, and in silico/machine learning (ML). Engineering techniques for altering non-enveloped virus tropism and their applications are summarized in Table 1 and explained in detail below.

#### 3.1.1. Rational Design

To develop a rational design scheme for altering AAV tropism, a comprehensive understanding of an AAV structure and its biology, especially an in-depth knowledge of AAV intracellular trafficking pathways in the target cells, are required. So far, two key approaches have been employed: mutational analysis and insertion of high-affinity peptides within the capsid protein sequence.

Early studies examined the AAV transduction efficiency and biodistribution using site-directed mutagenesis to identify the potential key amino acid residues on the capsid for binding to its target. For example, after altering a surface-exposed tyrosine on the AAV2 capsid, the central nervous system (CNS) transduction efficacy of AAV2 was enhanced significantly in the striatum and hippocampus of mice. The mutation of the capsid tyrosine residue helped AAV2 particles evade the ubiquitin–proteasome pathway and thus improved their intracellular trafficking to the nucleus for a better transgene expression [35,36]. In another example, an AAV8 variant whose surface-exposed tyrosine and threonine residues were substituted with phenylalanine and valine residues, respectively, showed improved brain transduction in the neonatal Mucopolysaccharidosis type IIIB (MPS IIIB) mouse, a disease model for a rare and devastating childhood disease caused by complete loss of function of the lysosomal hydrolase α-N-acetylglucosaminidase [37]. Today, enhanced modeling capabilities and mutagenizing tool kits allow for rapid screening of capsid amino acid contributions to AAV binding. The efficacy of transduction can also be improved by mutagenizing capsid surface proteins to evade the host immune response.

Another rational design strategy is to incorporate the specific ligand/peptide domains of known host-binding proteins into the exposed sites on AAV capsids. For example, in order to deliver therapeutic DNA into mitochondria efficiently, a 23-amino acid peptide with a mitochondria targeting sequence (MTS) was fused into the VP2 capsid protein of AAV2. The MTS-modified AAV redirected AAV particles to mitochondria and facilitated the delivery of the human NADH ubiquinone oxidoreductase subunit 4 (ND4) to the organelle. Expression of wildtype ND4 in a disease model led to restoration of defective ATP synthesis and suppressed visual loss and optic atrophy [38]. Another recent study utilized rational design to engineer and display a 6-mer peptide TVSALK on an AAV9 capsid. The resulting AAV9-derived capsid variant, AAV.CPP.16, showed an increased systemic gene delivery efficiency in both mice and nonhuman primates (NHPs). In addition, compared to the parental AAV9 vector, AAV.CPP.16 showed enhanced transcytosis via the blood–brain barrier (BBB) and improved transduction efficiency [39]. A combination of cell surface markers can be used to enhance the precision of in vivo gene delivery via receptor targeting. Dr. Buchholz’s research team (Goethe University, Frankfurt, Germany) designed ankyrin repeat proteins (DARPins) and mono- and bi-specific markers for CD4 and CD32a receptors on AAV2 capsids. The bi-specific DARPin-Targeted-AAVs (DART-AAVs) demonstrated much higher transduction efficiency in CD4/CD32a double-positive cells compared to that in the single-positive cells. This study demonstrated a novel strategy for high-precision gene delivery through tandem-binding regions on AAV capsids [40]. Some serotypes are more accommodating to modifications than others. For example, an AAVDJ VP1 capsid allows for large domain insertions with potential hotspots for AAV capsid engineering and redirecting AAV tropism [59]. Recently, this feature was used to insert a GFP-nanobody-binding domain into the VP3 capsid protein, and the modified capsid was successfully utilized in producing AAV virions and delivering its cargo. Rational design approaches are continuously improving due to increasing knowledge of AAVs’ structures and biologies.

#### 3.1.2. Directed Evolution

A directed evolution strategy to alter AAV tropism starts with the creation of a library of diverse AAV capsid variants, followed by selective pressure and screening of variants that exhibit desired properties. Typically, error-prone PCR or DNA shuffling are used to introduce random mutations into capsid genes [20]. For example, error-prone PCR was used for an in vivo directed evolution study on AAV2 capsids and resulted in a newly evolved variant, AAV2-retro, which can robustly travel retrograde in neuronal projections [41]. In another study, the capsid DNA from AAV serotypes 1–6, 8, and 9 were shuffled. The DNA fragments were recombined to create a chimeric AAV library. After several cycles of selection, a novel AAV variant capable of crossing the seizure-compromised blood–brain barrier (BBB) was identified [60]. Recently, researchers from Avimax Inc applied a new approach of artificial intelligence (AI)-guided directed evolution to identify a novel AAV capsid AAV2.N54. The new AAV capsid variant, AAV2.N54, exhibited improved tropism for mouse, pig, rabbit, and monkey retinas. AAV2.N54 delivery of vascular endothelial growth factor (VEGF)-Trap achieved therapeutic efficacy in mouse and rabbit disease models [42].

Peptide display is another advanced technique where peptides (typically randomized short amino acid 7-mers) are genetically fused or displayed on the surface of AAV capsids to alter tropism. This strategy has resulted in the discovery of novel central nervous system (CNS)-targeting capsids such as PHP.B [61], PHP.eB and PHP.S [43], BR1 [43] and AAV.BI30 [44]. Another resulting capsid from this strategy is AAV.CAP.B10, which contains an attenuated liver targeting MicroRNA (miRNA) and mediates brain-wide transgene expression in mice and marmosets [45].

At the present time, several evolution platforms have been successfully developed. The Cre-Recombination-based AAV Targeted Evolution (CREATE) was designed to develop AAV capsids that can more effectively transduce specific cell populations to express Cre recombinase in vivo. Briefly, this method uses the polymerase chain reaction (PCR) to introduce diversity into capsid gene fragments and to generate a capsid variant library. Then, the library is injected into Cre transgenic animals. Finally, the capsid sequences are selectively recovered from Cre+ cells to identify desired variants. The CREATE has successfully generated AAV variants, such as PHP.B [61], PHP.eB, and PHP.S [43], that efficiently and robustly transduce the adult mouse central nervous system (CNS) after systemic injection. Since AAV variants demonstrate altered tropism based on species and strains, this technique allows for selection of desired variants in targeted species/strains. Another systematic capsid evolution approach, called the Barcoded Rational AAV Vector Evolution (BRAVE), demonstrated efficient and large-scale selection of engineered capsids in a single screening round in vivo. Using the BRAVE approach, each virus particle displays a protein-derived peptide on the surface and carries a unique molecular barcode inside as part of its genome. The sequencing of RNA-expressed barcodes from a single-generation screening enables the selection of functional capsid structures. By using this method, several capsid variants were generated with specific properties, such as retrograde axonal transport and the retrograde infectivity of dopamine neurons in both rodent and human cells [46]. Another innovative platform for engineering AAV capsids is the Tropism Redirection of AAV by Cell-type-specific Expression of RNA (TRACER). This platform is based on the recovery of capsid library RNA transcribed from CNS-restricted promoters. The TRACER generated BBB-penetrating AAV variants with high efficiency in mouse brains. These variants demonstrated an up to 400-fold higher brain transduction over AAV9 following systemic administration [47]. Finally, researchers from Regenxbio Inc have developed a new capsid discovery engine, Novel AAV Vector Intelligent Guided Adaptation Through Evolution (NAVIGATE), to identify novel AAV3B and AAV8 variants with superior retina transduction profiles in multiple animal models. They have identified the novel AAV.PEPIN variant from multiple libraries that outperformed AAV8 in ocular tissue transduction via suprachoroidal space (SCS) administration in large animals [48].

#### 3.1.3. In Silico- or Machine Learning (ML)-Based Design

The in silico design of AAV capsids integrates computational modeling, bioinformatics, algorithms, and machine learning techniques to predict and optimize the development of novel capsids with improved characteristics. AAV capsids have diverse properties that originate from their evolutionary paths. Ancestral reconstruction algorithms can be used to create novel capsid variants [49,62]. One such algorithm has resulted in the discovery of a new AAV capsid variant, Anc80L65, with improved thermostability and a comparable production yield. Anc80L65 robustly delivers genes to murine livers, muscles, inner ears, retinas, and kidneys [49,50,51,52,53] and NHP livers and muscles [49].

Machine learning (ML) is a branch of artificial intelligence (AI) used to develop algorithms and models to guide AI in predicting and developing new algorithms based on data. The first step would be to generate an informative training dataset including positive and negative controls. A pioneering research project by George Church, Ph.D. (Massachusetts Institute of Technology, Cambridge, MA, USA) has generated an AAV2 capsid fitness landscape with characterized single-codon substitutions, insertions, and deletions of all capsid amino acids, as it relates to in vivo gene delivery. These landscapes are largely enabled machine-guided designs [54]. Deep learning is another method used to design highly diverse AAV2 capsid variants by the research team of Eric Kelsic, Ph.D. (Harvard Medical School, Boston, MA, USA). They have demonstrated that, even when trained on limited data, deep neural network models accurately predict capsid viability across diverse AAV capsid variants [55]. Currently, most researchers engaged in ML-assisted capsid engineering focus on improving particle production and immune evasion [49,54,55,56].

### 3.2. Adenoviruses

Adenoviruses are double-stranded DNA viruses enclosed in an icosahedral nucleocapsid that are capable of infecting a broad range of cell types, including both dividing and non-dividing cells. This versatility makes them effective vectors for gene delivery, offering high transduction efficiency, a large cargo capacity (~7.5–8 kb of foreign DNA), and the ability to generate high-titer viral stocks [63]. These vectors are particularly advantageous for delivering genes intended for short-term expression, such as in cancer immunotherapy, vectored vaccines, or scenarios where transient gene expression suffices.

In cancer therapy, adenoviral oncolytic viruses are engineered to selectively target and destroy cancer cells while delivering genes that either stimulate an immune response against the tumor or introduce “suicide genes” that convert non-toxic substances into toxic compounds, effectively killing the cancer cells [64,65]. However, adenoviral vectors can provoke strong immune responses, particularly when administered intravascularly, potentially leading to cytokine storms, hepatotoxicity, or thrombocytopenia, which may necessitate the use of immunosuppressive strategies in some gene therapy applications [66].

In the context of vector-based vaccines, adenoviruses are used to deliver pathogen-derived antigens to elicit robust immune responses, as demonstrated in the development of COVID-19 vaccines, such as the AstraZeneca vaccine [67].

For most wildtype adenoviruses, tropism is primarily determined by the interaction between the virus and the coxsackievirus and adenovirus receptor (CAR) on host-cell surfaces. Capsid proteins, including the fiber and penton base, play crucial roles in mediating viral entry into host-cells [68]. Over the last few decades, several strategies have been developed to engineer CAR-independent entry by creating recombinant adenoviral capsid proteins through the addition of targeting peptides to their capsids (added to the fiber knob domain, fiber shaft, penton base, pIX, or hexon), by utilizing fiber–penton base chimeras or the capsid proteins of another adenovirus with tropism toward another species (fiber pseudotyping), or through polymer-coating to create stealth vectors [57].

One of the latest advancements in adenovirus-based vector engineering is the development of artificial vectors for intravascular delivery (AVIDs). These vectors are designed to deliver genes to human hematopoietic stem and progenitor cells (HSPCs) in vivo [58]. Given that intravascular delivery of adenoviruses typically triggers strong immune responses, which can impede gene delivery to bone marrow-resident cells, AVIDs have been engineered to target cells expressing CD46 or DSG2 receptors and Laminin-interacting integrins (α6β1, α6β4, α3β1, and α7β1). The use of cell-type-specific promoters further restricts transgene expression to the target cells, resulting in a safe and efficient platform for gene delivery to human bone marrow cells. The combined strategy has created a safe and efficient platform for gene delivery to human bone marrow cells [58].

## 4. Engineering Tropism of Recombinant Enveloped Viruses (Retro/Lentiviruses, Rabies-dG and HSVs)

Enveloped viruses are characterized by a lipid bilayer that surrounds their capsids and are embedded with viral glycoproteins that bind specific receptors on the host-cell surface to mediate entry. One of the most common methods for engineering the tropism of enveloped viruses is pseudotyping, where the envelope glycoprotein of one virus is replaced by the envelope glycoprotein of another [69]. Viruses can bud from different locations on their host-cells, and the site of this budding is an important factor in assessing the envelope compatibility between enveloped viruses. Furthermore, the site of virus budding can influence host immune response and viral pathogenesis.

### 4.1. Budding from the Plasma Membrane vs. Intracellular Vesicles

The process of viral budding, where newly formed virus particles are released from the host, differs based on a virus’s characteristics. For example, the Influenza virus and HIV viral envelopes and matrix proteins are transported and assembled on the host plasma membrane. Later, other viral components and their genomes (RNA) are recruited to these assemblies and the host-cell plasma membrane is pinched off to release new virions [69]. During this process, the host lipid membrane is incorporated into the viral assembly. Other viruses such as coronaviruses (SARS-CoV-2) and Herpesviruses are assembled on the intracellular membranes of Golgi and endosomes and bud into vesicles that are transported and fused to the host plasma membrane to release the newly formed virions [70]. Viruses budding from the vesicles acquire envelope lipids from intracellular components. Understanding these differences in budding processes helps in engineering envelopes for pseudotyping recombinant enveloped viruses.

### 4.2. Pseudotyping Envelopped Viruses

The following sections will describe the process of viral pseudotyping and the applications of the engineered tropism for enveloped viruses (summarized in Table 2).

#### 4.2.1. Retro/Lentiviral Pseudotyping

Lentiviruses and gamma-retroviruses (hereinafter referred to as retroviruses), both members of the Retroviridae family, are enveloped viruses with a genetic cargo of two positive-sense single-stranded RNAs. Upon infecting a host-cell, their RNA genome is reverse-transcribed into DNA, which is then randomly integrated into the host genome for stable expression [101]. Gamma-retroviruses are limited to infecting dividing cells, making them a useful research tool for studying cell fate and neuronal development. In contrast, lentiviruses can infect both dividing and non-dividing cells, making them a suitable vector for gene delivery to a myriad of in vivo and in vitro cell types. The random integration of retro/lentiviral genomic cargo can result in mutations that affect gene expression and cell function [102]. To minimize the risk of insertional mutation, researchers have developed integrase deficient recombinant lentiviruses (IDLVs) that are efficient and non-integrative vectors [102]. Both recombinant retro and lentiviruses are capable of accommodating large insert sizes (up to 9 kb). They are among the most popular and versatile tools for gene delivery and cell therapy [101].

The entry of retro/lentiviruses into cells is mediated by envelope glycoproteins. By substituting the native envelope protein with those from various other viruses, researchers can extend the range of target cells or achieve specific targeting beyond the interaction of the gp120 envelope protein with CD4 on human lymphocytes. A commonly used envelope protein, the Vesicular Stomatitis Virus G Protein (VSV-G), binds to the low-density lipoprotein receptor (LDL-R), which is present on most mammalian cell types. Pseudotyping with VSV-G broadens the range of cells that can be targeted. Alternatively, other viral envelope proteins can be employed for more precise targeting of specific cell types [69].

Psuedotyping can also be employed to create in vitro models to safely study the viral proteins of hazardous viruses. For example, during the COVID-19 pandemic, a number of laboratories employed recombinant lentiviruses pseudotyped with SARS-CoV2 variants to study the virus spike protein structure and screen antiviral agents in a Biosafety Level 2 (BSL2) laboratory environment (Figure 2).

Lentiviruses are particularly advantageous for gene therapy due to their high transduction efficiency and ability to provide stable, long-term expression of transgenes. Recent applications include treating certain blood cancers by modifying host T-cells. These immune cells, which target and destroy foreign entities like viruses, bacteria, and cancer cells, can be engineered to overcome the mechanisms that cancer cells use to evade detection. By introducing a gene that encodes chimeric antigen receptors (CARs), T-cells are enabled to recognize and bind to cancer cells. These CAR-T cells are then reintroduced into the patient to target and eliminate the cancer cells [103]. This approach has been effective in treating B-cell malignancies and multiple myeloma, and it has received FDA approval [104]. Research is also exploring CAR-T cell therapy for solid tumors, though its effectiveness in this area is still under investigation [104]. Moreover, CAR-T cell therapy is being adapted for autoimmune diseases. The INT2106, also known as Gen 2.1 Fusogen, uses pseudotyped lentiviruses with a CD7 binder protein and a modified VSV-G to improve the above therapy. The CD7 binding enables the lentiviral particles to target CD7+ cells, such as T-cells and NK cells, while the modified VSV-G protein, with altered residues, prevents binding to LDL-R. Developers introduced negatively charged side chains at specific sites in VSV-G to destabilize critical ionic interactions with LDL-R [105]. The modified VSV-G protein maintains a pH-dependent fusion but avoids binding to the LDL-R, limiting the targeted cell types to CD7+ cells [106]. This targeted delivery enables the generation of functional CAR T and NK cells expressing the CAR19 transgene, which specifically eliminates CD19+ B cells after a single intravenous injection of the INT2106 lentivirus. This method allows for the production of functional CAR-T and NK cells that specifically target CD19+ B cells with a single intravenous injection, simplifying treatment and enhancing patient accessibility.

Pseudotyped lentiviruses are also utilized to transduce hematopoietic stem cells and neural stem cells. While VSV-G-pseudotyped lentiviruses are favored for their broad tropism and stability, they are less effective in transducing quiescent hematopoietic stem cells (HSCs) and other primary cells [71,72]. To address this limitation, researchers are exploring alternative envelope proteins from viruses such as the Baboon Endogenous Retrovirus (BaEV), Nipah Virus (NiV), and Sendai Virus (SeV) [72,73,74]. These alternative envelope proteins aim to enhance the specificity and efficiency of gene delivery, paving the way for more effective treatments for various genetic diseases and conditions.

The brain remains one of the most enigmatic organs in the human body, with the intricacies of neuronal circuits and cellular interactions not yet being fully understood. In neuroscience, pseudotyped lentiviruses have become invaluable tools for studying the brain’s complex network. Pseudotyped lentiviruses incorporating the Rabies virus envelop protein G (RV-G) or fusion envelope glycoproteins comprised of RV-G and VSV-G segments, termed highly efficient retrograde gene transfer (HiRet) and neuron-specific retrograde gene transfer (NeuRet), have gained widespread use in neuroscience for studying neural structure. These biocehmical tools have been used to deliver tracers, sensors, and actuators to mammalian brains to identify neural network connections [107,108]. RV-G or HiRet/NeuRet pseudotyped lentiviruses enable efficient and specific gene delivery to neurons by exploiting their retrograde transport mechanisms.

Additionally, fusion proteins combining VSV-G with avian sarcoma leucosis virus envelope proteins EnvA or EnvB, along with their corresponding receptors TVA and TVB, enable precise targeting of specific neuronal subsets or even single neurons for functional analysis. This approach could employ a two-vector delivery system where AAV vectors are used to deliver TVA or TVB receptors to target neurons. Alternatively, neuronal-specific TVA/TVB knock-in animal models can be used for targeting Env A/B pseudotyped lentiviruses [75].

Astrocytes, the predominant glial cells in the central nervous system (CNS), play crucial roles in various physiological processes, including neurotransmitter clearance, blood–brain barrier stabilization, and synapse formation. Researchers have utilized lentiviruses pseudotyped with envelope glycoproteins from the lymphocytic choriomeningitis virus (LCMV) or Moloney murine leukemia virus (MuLV) to selectively transduce astrocytes in the rat substantia nigra [109]. Mokola-pseudotyped lentiviruses also show a preference for astrocytes compared to VSV-G-pseudotyped vectors [110]. Furthermore, lentiviruses pseudotyped with a modified Sindbis envelope displaying anti-GLAST IgG exhibit preferential targeting of astrocytes both in vitro and in vivo [111]. Recent advances include a second-generation pseudotyped lentivirus vector specifically targeting astrocytes for efficient gene expression or silencing [76]. These innovations enhance the potential for targeted gene therapy within the brain.

Pseudotyped lentiviruses are also pivotal in infectious disease research, supporting vaccine development and antiviral drug discovery. Lentiviral vectors pseudotyped with various viral glycoproteins have been used to develop vaccines against Influenza, HIV, SARS-CoV-2, Ebola, and malaria. This approach provides safety advantages over traditional attenuated vaccines, efficient antigen delivery, and robust immune responses. For example, lentiviruses pseudotyped with the SARS-CoV-2 spike protein have been instrumental in evaluating antibody neutralization and studying antibody persistence following vaccination [77,78,79,80]. Since the SARS-Cov-2 virus buds from host vesicles, a modified version of the viral spike protein was utilized for pseudotyping lentiviruses. The deletion of the last 19 amino acids of the viral spike protein redirected it from the endoplasmic reticulum (ER) to plasma membrane for the efficient pseudotyping of recombinant lentiviruses [112]. These pseudotyped lentiviruses are also used to screen antiviral drugs and investigate SARS-CoV-2 infection mechanisms, potentially revealing new therapeutic targets [113].

#### 4.2.2. Rabies-dG and VSV-dG Psueodtyping

Indiana vesiculovirus, formerly the Vesicular Stomatitis Indiana virus (VSIV or VSV), and the Rabies lyssavirus virus (Rabies virus) are members of the family Rhabdoviridae that have a single-stranded, negative-sense RNA genomes [114]. The assembly of the VSV and Rabies viruses occur at the plasma membrane. Therefore, techniques used for pseudotyping retro/lentviral viruses described in the previous section can be applied to pseudotyping glycoprotein-deleted VSV and Rabies (respectively referred to as rVSV-dG and Rabies-dG). These recombinant viruses offer robust expression, but their use is limited since they cannot accommodate cell specific promoters or transcriptional regulators (e.g., Cre-lox sequences).

The Recombinant Vesicular Stomatitis Virus (rVSV) has been used for vaccine development because it induces potent innate and adaptive immune responses [115]. Shortly after the start of the COVID-19 pandemic, SARS-CoV-2 spike pseudotyped rVSV-dG were employed for structural studies of the spike and therapeutic drug development [93]. Since the wildtype VSV envelope glycoprotein (VSV-G) has a broad host range, it is often used for pseudotyping other viruses (mentioned in Section 4.2.1).

The neurotropic recombinant Rabies-dG virus is an excellent tool for tracing neuronal circuits [94]. Rabies-dG is often used in conjunction with a helper virus in a two-step system to ensure safety and target specificity. Similar to lentiviruses, the envelope glycoprotein of avian sarcoma and leukosis virus (ASLV) with subgroups A, B, E, etc. can be used to pseudotype Rabies-dG. In the two-step approach, a helper virus is first utilized to deliver TVA/TVB/TVE, etc. to allow the pseudotyped Rabies-dG to enter the cell. This helper virus often also carries the Rabies glycoprotein gene that is necessary for the retrograde transmission of the virus. Once the genes delivered by the helper virus are expressed, neurons are infected with the pseudotyped Rabies-dG for monosynaptic tracing [94]. The recombinant Rabies-dG virus is engineered to be replication-deficient or attenuated, reducing the risk of viral replication and toxicity. Psuedotyped recombinant Rabies-dG has provided neuroscientists with a powerful tool for studying neuronal connectivity, neural circuits, and the functional organization of brains with high precision [116].

#### 4.2.3. HSV Pseudotyping and Its Applications

Herpes simplex virus type 1 (HSV-1) is a versatile tool in gene therapy, vaccine development, and research due to its broad cell tropism, infecting a wide variety of cell types. Its two main vector forms, replication-defective HSV-1 and HSV-1-based amplicons, offer advantages such as large cloning capacities (30kb and 150kb, respectively), high transduction efficiency, and reduced cytotoxicity, especially with the amplicon system [95,117,118]. HSV-1 pseudotyping provides additional control by incorporating envelope proteins from other viruses. A prime example is Vesicular Stomatitis Virus glycoprotein G (VSV-G), which can replace HSV-1 glycoprotein D (gD) [119]. This offers potential benefits: (1) Enhanced gene transfer: VSV-G might improve the efficiency of delivering genes into target cells [119]. (2) Studying HSV-1 entry: scientists can use VSV-G pseudotyping to investigate how HSV-1 enters cells [120,121]. While studies incorporating VSV-G or chimeric VSV-G/HSV proteins achieved high incorporation rates in gD-deficient HSV-1 mutants, only native VSV-G mediated infection [119]. This highlights the concern of maintaining functionality during envelope modification. In contrast to VSV-G pseudotyping, HSV-1 envelope proteins could be harnessed for the selective targeting of specific cell types, such as T cells, B cells, hepatocytes, skeletal muscle cells, or neurons. This approach, already demonstrated with other viral vectors like lentiviruses, can minimize off-target effects and enhance therapeutic specificity [122,123].

Based on evidence from other envelope pseudotyped viruses in vaccine development [98,99,100], HSV-1 pseudotyping may also offer a promising path for vaccine development beyond gene therapy applications. By incorporating envelope proteins from diverse viruses like HIV, Nipah, Rabies, SARS-CoV-2, or Ebola, HSV-1 can be engineered as a multivalent vaccine candidate, potentially offering protection against multiple pathogens simultaneously [95,96,97]. Utilizing pseudotyped replication-defective HSV vectors also presents a safer alternative to traditional live attenuated vaccines [95]. However, challenges remain, including optimizing the incorporation efficiency of foreign proteins into HSV particles, mitigating the impact of host immune responses to the vector itself, and ensuring the safety of pseudotyped viral vectors through rigorous testing before clinical use [96,124]. Despite these hurdles, HSV-1 pseudotyping remains a powerful tool with wide-ranging applications in gene therapy, vaccine development, and virology research. As knowledge of viruses and vector design progresses, the potential applications of pseudotyped HSV vectors are expected to expand, paving the way for novel disease treatments and further insights into viral pathogens.

### 4.3. Chimera Envelope Proteins for Viral Vector Pseudotyping

Chimera envelope proteins, engineered fusion proteins, are a versatile tool in gene therapy. They are used to pseudotype viral vectors, altering their tropism and improving their transduction efficiency. These proteins consist of two main parts: portions of a viral protein necessary for incorporation into the virion and sequences designed to interact with specific host-cell proteins. By modifying these components, researchers can expand or limit the range of cells targeted by a gene therapy vector, improve its stability and transduction efficiency, and tailor the vector’s tropism for precise gene delivery [85,86]. Over the past decades, various targeting molecules, including short peptides, ligands, and even single-chain antibody variable fragments (scFvs), have been incorporated into viral envelope proteins to achieve cell-type-specific targeting. Ligands like insulin-like growth factor I (IGF-I), EGF, erythropoietin (EPO), and stromal-derived factor-1α (SDF-1α) have been inserted into the N-terminal region or the receptor-binding domain of envelope proteins [87,88,89,90]. While this technique shows great promise, challenges remain. Peptide insertion can disrupt the structure and function of the viral vector, and the high-throughput screening is necessary to ensure the desired outcome [125]. Another approach involves combining fragments from different viral envelope proteins. For instance, chimeric GALV-Env proteins, derived by incorporating internal domains from murine leukemia viruses (MLVs) into the Gibbon Ape Leukemia Virus envelope protein (GALV-Env), have been used to pseudotype lentiviral vectors [91]. A recent development, coGALV-Env, a codon-optimized variant of GALV-C4070A, has shown even greater efficiency in pseudotyping and offers a cost advantage in large-scale production [92]. Overall, chimera envelope proteins provide a powerful tool to optimize viral vectors for gene therapy, combining the strengths of different viral envelopes to achieve the desired targeting, stability, and efficiency.

## 5. Discussion and Conclusions

Engineering the tropism of viral vectors is a dynamic area of research with numerous advancements. Viruses are powerful vehicles for gene delivery that can be engineered to target specific cell types, tissues, or species. There are currently hundreds of AAV serotypes and variants, with wildtypes, mutants, and chimeric capsids being developed for research and delivering therapeutics. However, there are still some major limitations and concerns that need to be addressed in the future studies. For example, AAV-PhP.B was selected in mouse strain B6/C57. It indeed demonstrated high transduction efficiency in the central nervous system (CNS) of B6/C57 mice after intravenous or retro-orbital injections. However, the result did not apply to other mouse strains or species, such as BALB/cJ mice or nonhuman primates [61,126]. To improve the engineered vector applications and cross-species translatability, researchers must screen and test a candidate vector in multiple models, especially in animal models close to humans, to develop therapeutics. Currently, AAV transduction specificity is mainly determined at the tissue level by vector tropism and regulatory elements such as promoters and microRNA (miRNA) target sequences [127]. In the future, capsid engineering may focus on the tropism within a cell, for example, targeting organelles and mitochondria in cells.

Another concern of AAV capsid engineering is how to keep a balance between altering or improving AAV properties and preserving virus integrity. Sometimes, AAV capsid modification improves a trait at the expense of AAV packaging efficiency. Recently, the Deverman group developed a machine learning-guided approach, Fit4Function, that can simultaneously screen 7-mer-modified-AAV9 capsid sequences to match multiple traits essential for therapeutic vectors [128]. In the future, the very promising design tool, machine learning-guided multi-trait capsid identification, may not only reduce the labor of the conventional selection of AAV capsid libraries but may also be able to identify novel AAV capsids by possessing multiple traits simultaneously (e.g., specificity, manufacturability, and low immunogenicity).

Viral vector tropism is often utilized for gene delivery to specific cells or tissues. However, tropism could also be engineered to induce an immune response to an antigen displayed on the surface of the virus. Use of adenoviral-based vectors to display pathogen-derived antigens has greatly improved the production of vaccines to fight infections. CAR-T-focused therapeutics have offered promising cancer treatments. The ongoing protein engineering research and designs must address several gaps and challenges in capsid and envelope protein engineering.

The complexity of virus–host receptor binding and the detailed mechanisms by which enveloped viruses interact with their receptors are often not fully understood. Detailed host receptor structures and comprehensive pathway maps of virus entry and assembly are needed to better engineer tropism. Some viruses may bind to multiple receptors or use alternative entry pathways, which complicates engineering efforts. Another challenge in targeting vectors is to develop better computational tools for predicting tropism and designing experimental models for validating the constructs. Each altered tropism has the potential to affect virus stability or altered immunogenicity. Engineering viruses to evade the host immune system remains a significant challenge.

While changes in envelope proteins can help in altering vector targets and controlling immune responses to engineered viruses, safety and off-targets effects are significant factors to consider. Ensuring that engineered viruses do not cause adverse effects or lead to unintended consequences is crucial.

Engineering virus envelope or capsid proteins has provided compelling evidence for controlling virus tropism. Recent studies have also shown that modulating the expression of co-receptors, such as tetraspanins, can alter virus tropism or enhance viral vector transduction efficiency. Tetraspanins, a family of transmembrane glycoproteins with 33 members identified in humans, have emerged as crucial players in viral infections, influencing multiple stages of the viral life cycle. Notably, CD9, CD63, CD81, CD82, CD151, and TSPAN7 (CD231) have been implicated in numerous infectious disease pathologies [129,130]. Tetraspanins are involved in the pathogenesis of both non-enveloped (e.g., human papillomavirus) and enveloped (e.g., HIV, Zika, Influenza A virus, and coronaviruses) viruses. Their influence extends through multiple stages of infection, from initial cell membrane attachment to syncytium formation and viral particle release. Introducing sequences on AAV capsids, which are recognized by cell type-specific receptors or co-receptors, has been developed as a strategy for altering tropism. For example, tetraspanin CD9 overexpression led to enhanced AAV production and transduction [131], implying the potential role of tetraspanins, as co-receptors, in the future development of AAV capsid engineering.

Previous studies have demonstrated that virus particles containing CD9 molecules on lentiviral envelope protein (LV-VSVG-CD9) significantly increased gene delivery efficiency compared to the VSV-G-pseudotyped lentivirus (LV-VSVG) in numerous cell lines, including HEK293, HeLa, SH-SY5Y, and B and T lymphocytes. Conversely, cells with high CD9 expression levels exhibit better transduction efficiency than cells lacking CD9 [132]. Therefore, co-receptors like tetraspanins, with their extensive involvement in various viral infections, represent promising targets for gene delivery and antiviral therapies. Ongoing research is actively exploring their potential in enhancing gene transduction efficiency and combating a broad spectrum of viral diseases.

Research into engineering the tropism of enveloped viruses continues to advance, but these gaps highlight areas where further work is needed. While pseudotyped lenti/retroviruses are powerful tools for gene therapy, moving from research to clinical applications is hampered by numerous challenges. One major challenge is the severity of the host immune response against pseudotyped viral particles that could limit the therapeutic efficacy and potentially lead to adverse outcomes. Additionally, the pseudotyped lenti/retroviral vectors can randomly integrate their genomic DNA into their host chromosomes, increasing the possibility of gene interruption, transcriptional dysregulation, or the risk of developing cancer. The off-target effects on healthy cells that are inadvertently transduced or affected by the virus-based therapy also pose concerns. Furthermore, lentivirus used for T cell engineering (i.e., CAR-T therapy) is limited by the patient’s disease state and/or progress since there may not be adequate T cell numbers for effective treatment. Moreover, this type of therapy is currently only used in therapy for blood cancer, not solid tumors. Despite these challenges, lentiviruses are effective gene delivery vehicles, and further research to develop better targeting capabilities and reduce host immune responses will open doors for the safe use of lentiviruses in future gene therapy trials.

As mentioned before, improving virus capabilities sometimes will add to production challenges. Pseudotyped HSV faces challenges like lower viral yields compared to wildtype strains, impacting production efficiency. Additionally, the incorporation rate of foreign glycoproteins, while possible, is often lower than native ones, further reducing pseudotyped virus yields. Moreover, HSV-pseudotyped viruses typically show reduced infectivity, with examples like VSV-G-pseudotyped HSV demonstrating a significant decrease in plaque formation compared to wildtype HSV. These limitations highlight the need for further optimization when using HSV-pseudotyped viruses in research for studying viral entry, developing vaccines, and evaluating antiviral therapies. Addressing these challenges requires multidisciplinary approaches, including improved computational tools, better experimental models, enhanced understanding of immune interactions, and rigorous safety assessments. As the field progresses, overcoming these gaps will be critical for the development of effective and safe viral vectors for gene therapy, vaccine development, and other applications.

## Figures and Tables

**Figure 1 ijms-25-11094-f001:**
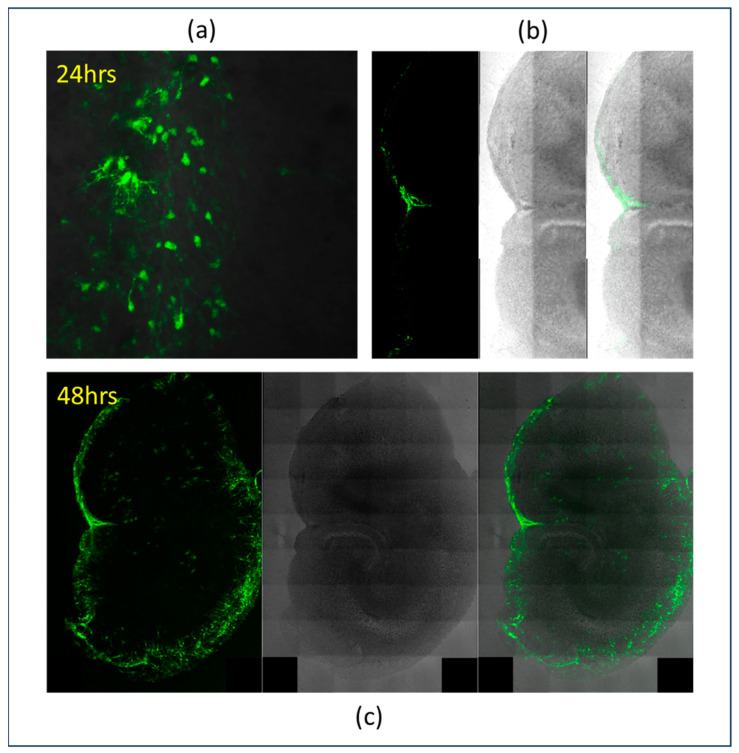
Cultured mouse brain slices, prepared as described previously [11], were transduced with baculovirus, delivering a green fluorescence protein (GFP) pseudotyped with Vesicular Stomatitis Virus (VSV) glycoprotein (BacMam technology, Thermo Fisher Scientific). (**a**) After a period of 24 h following transduction with the engineered baculovirus, a GFP is expressed in the vicinity of the infection area (20× objective, zoom of 1, single tile). (**b**) Images of the mouse brain slice, depicting the scale of gene expression with fluorescence only, visible light, and the overlap of both fluorescence and visible light (20× objective, zoom of 1, assembled tile scans). (**c**) Robust GFP expression in brain slices are evident 48 h post-infection—images depicting fluorescence only, visible light, and the overlap of both fluorescence and visible light. Black boxes in (**c**) are regions where no data was acquired by tile scan (20× objective, zoom of 1, assembled tile scans).

**Figure 2 ijms-25-11094-f002:**
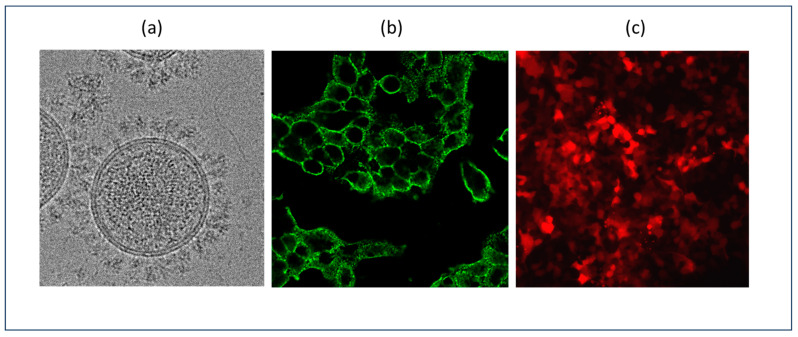
Recombinant lentiviruses pseudotyped with SARS-CoV-2 Spike delivering a Red Fluorescent Protein (Lenti-RFP-Spike). (**a**) Cryo-EM image of the purified Lenti-RFP-Spike imaged on Talos Arctica 200kV equipped with Gatan K2 Summit detector at a magnification of 36,000×. (**b**) HEK293T stably expressing human Angiotensin-Converting Enzyme 2 (HEK293T-hACE2) infected with a Lenti-RFP-Spike, fixed with 1% formaldehyde and stained with anti-spike antibodies conjugated to GFP (20× objective, zoom of 2, 400× magnification). (**c**) Lenti-RFP-Spike binds to the hACE2 receptor on the surface of the HEK293T-hACE2 cells to enter and express RFP in infected cells (20× objective, 40× magnification).

**Table 1 ijms-25-11094-t001:** Engineering Tropism of Non-Enveloped Viruses.

Methods	Approach	Applications/Examples	Ref.
Rational Design	Mutation of surface-exposed tyrosine and threonine	Improved AAV2 and AAV8 transduction in CNS.	[35,36,37]
	Ligand/peptide incorporation into the exposed sites on AAV capsids	The MTS-modified AAV2 redirected AAV particles to the mitochondria. TVSALK on AAV9 improved systemic gene delivery efficiency via BBB. DARPins markers on AAV2 improved the transduction efficiency in CD4/CD32a double-positive cells	[38,39,40]
Directed Evolution	Error-prone PCR and DNA shuffling	The evolved AAV2-retro robustly travel retrograde in neuronal projections. AAV2.N54 exhibited an improved tropism for mouse, pig, rabbit, and monkey retinas.	[41,42]
	Peptide display	Novel CNS-targeting capsids, PHP.B, AAV.BI30, andAAV.CAP.B10.	[43,44,45]
	CREATE	Novel CNS-targeting capsids, PHP.B, PHP.eB, PHP.S.	[43]
	BRAVE	Novel AAV variant with retrograde transport and infectivity of dopamine neurons in both rodent and human cells.	[46]
	TRACER	BBB-penetrating AAV variants with high efficiency in mouse brain.	[47]
	NAVIGATE	Novel AAV3B and AAV.PEPIN variants with superior retina and ocular transduction profiles in multiple animal models.	[48]
In silico- or ML-based Design	Ancestral reconstruction algorithms	Novel Anc80L65 variant with improved thermostability and delivery efficiency.	[49,50,51,52,53]
	Machine learning	Improving AAV production and immune evasion.	[49,54,55,56]
Genetic Modification	Incorporation of targeting peptide sequences	Engineer CAR-independent entry.	[57]
	Development of artificial vectors	Artificial vectors for intravascular delivery (AVIDs) for gene delivery to human hematopoietic stem and progenitor cells	[58]

**Table 2 ijms-25-11094-t002:** Engineered Tropism of Pseudotyped Enveloped Viruses and Their Applications.

Virus	Pseudotyping	Tropism	Applications	Ref.
Retro/ lentivirus	VSV-G	Broad (LDL-R positive cells)	Gene delivery, creation of stable-cell lines	[71,72]
	(1) BaEV	(1) CD34 positive SC	Gene delivery to cells with less efficient VSV-G-pseudotyped lentiviruses transduction	[73]
	(2) NiV	(2) Ephrin B2 positive cells		[72]
	(3) SeV	(3) Hematopoietic SC		[74]
	ASLV (e.g., EnvA/EnvB)	Neurons expressing corresponding receptors. (e.g., TVA/TVB)	Functional analysis or synaptic tracing of neurons	[75]
	LCMV and MuLV	Brain cells, especially astrocytes	Efficient gene delivery to astrocytes	[76]
	(1) SARS-CoV2 spike	(1) Host-cells for SARS-CoV2 (ACE2-positive)	Transduce host-cells expressing corresponding viral receptors for vaccine development and antiviral drug discovery	[77,78,79,80]
	(2) Ebola envelope glycoprotein	(2) TIM-1 positive cells		[81,82]
	(3) Influenza hemagglutinin	(3) sialic acid receptors expressing cells		[83,84]
	Chimera envelopes:		Notably, (1) and (2) enhance precise gene delivery. Additionally, (3) mediates efficient transduction of B and T cells, improves virus particle stability, and increases virus production	
	(1) Growth factors (e.g., IGF-I, EGF, EPO, SDF-1α)	(1) Cells expressing corresponding receptors		[85,86]
	(2) Single-chain antibody variable fragments (scFvs)	(2) Cells expressing targeted epitopes		[87,88,89,90]
	(3) Combined fragments from different viral envelops (e.g., GALV-Env and GALV-C4070A)	(3) B and T cells		[91,92]
VSV-dG	SARS-CoV-2 spike	Host-cells for SARS-CoV-2 (ACE2-positive)	Structural studies of the spike protein and therapeutic drug development	[93]
Rabies-dG	ASLV (e.g., EnvA/EnvB/EnvE)	Neurons expressing corresponding receptors (e.g., TVA/TVB/TBE)	Neural circuit tracing.	[94]
HSV	VSV-G	Broad (LDL-R positive cells)	Virus pathogenesis research	[95,96,97]
	HIV, Nipah, Rabies, SARS-CoV-2, Ebola envelope proteins	Antigen presenting cells	Vaccines and antiviral drugs development	Based on evidence from other enveloped viruses [98,99,100]

Abbreviations: Stem Cells (SCs); Angiotensin-converting enzyme 2 (ACE2); VSV-G: Vesicular Stomatitis Virus G Protein; LDL-R: low-density lipoprotein receptor; BaEV: Baboon Endogenous Retrovirus; NiV: Nipah Virus; SeV: Sendai Virus; ASLV: avian sarcoma and leukosis virus; Env: envelope; LCMV: lymphocytic choriomeningitis virus; MuLV: Moloney murine leukemia virus; SARS-CoV2: Severe acute respiratory syndrome coronavirus 2; ACE 2: Angiotensin-converting enzyme 2; TIM-1: T-cell immunoglobulin and mucin 1, VSV-dG: Glycoprotein-deleted Vesicular Stomatitis Indiana virus; Rabies dG: Glycoprotein-deleted Rabies lyssavirus virus; HIV: Human immunodeficiency virus.

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
