# Peer review of "Molecular Engineering of Virus Tropism"

_ijms, 2024, doi:10.3390/ijms252011094_

Round 1

Reviewer 1 Report

Comments and Suggestions for Authors

This manuscript gives a detailed overview of the current research progress on engineered viral vectors. It covers both enveloped viruses (such as Retroviruses, Lentiviruses, and Rabies viruses) and non-enveloped viruses (like AAV and Adenoviruses). The manuscript thoroughly discusses the various efforts made to enhance the tropism of engineered viruses and highlights the achievements resulting from these efforts.

The overall content of the manuscript is solid, but some sections could still be streamlined for clarity. For example, Line 131 to 134, while easy to understand, are somewhat verbose and could benefit from a more concise revision.

In Table 2, the comma at the end of "(A) BaEV" can be removed.

The conclusion and discussion sections provide a comprehensive summary of the manuscript's content. However, adding a discussion about the limitations of these engineered viruses in practical applications, along with insights into future research directions, would enhance the depth of these sections.

Author Response

Comments and Suggestions for Authors

This manuscript gives a detailed overview of the current research progress on engineered viral vectors. It covers both enveloped viruses (such as Retroviruses, Lentiviruses, and Rabies viruses) and non-enveloped viruses (like AAV and Adenoviruses). The manuscript thoroughly discusses the various efforts made to enhance the tropism of engineered viruses and highlights the achievements resulting from these efforts.

Comment 1: The overall content of the manuscript is solid, but some sections could still be streamlined for clarity. For example, Line 131 to 134, while easy to understand, are somewhat verbose and could benefit from a more concise revision.

Response 1: Thank you for your comment. We agree with you and lines 131-134 were re-written to be more concise and highlighted in yellow. The new line numbers are 148-150.

Comment 2: In Table 2, the comma at the end of "(A) BaEV" can be removed.

Response 2: Thank you for noticing. This was a typo and the comma after BaEV was remove.

Comment 3: The conclusion and discussion sections provide a comprehensive summary of the manuscript's content. However, adding a discussion about the limitations of these engineered viruses in practical applications, along with insights into future research directions, would enhance the depth of these sections.

Response 3: This is a great and insightful suggestion. We thank the reviewer for the suggestion. We have added several highlighted sections to the discussion section in lines from 604-629 and 647-700 to address possible concerns and limitations of the engineered tropism and offer possible future directions in developing new strategies. Reviewer’s comment has definitely helped us improve the manuscript.   

Reviewer 2 Report

Comments and Suggestions for Authors

The review article refers to the molecular engineering of virus tropism. It has an adequate rationale, exposing the different elements relating to capsids and proteins of various viruses and possible targets. Even though the article is well-written, some aspects are missing. One is the relationship between receptors and coreceptors. The specific binding to receptors is partially functional, especially in neurological cells. Is there any advantage of introducing sequences recognized by some specific receptors on the cell surface to specific cell types? For example, how functional can the interaction of virus receptors glycoproteins be? These glycoprotein receptors, such as tetraspanins, can be associated with receptors inducing internalization.  There are only two tables and no figures, which, for a review paper of this dimension, is not adequate. At least two figures should be included, including constructs and interactions of the constructs. Part of the text should also be checked since there are some similarities with previously published research. Finally, what are the possible limitations? Is there any certainty of safe molecular modifications?

Author Response

Comment 1: Even though the article is well-written, some aspects are missing. One is the relationship between receptors and coreceptors. The specific binding to receptors is partially functional, especially in neurological cells. Is there any advantage of introducing sequences recognized by some specific receptors on the cell surface to specific cell types? For example, how functional can the interaction of virus receptors glycoproteins be? These glycoprotein receptors, such as tetraspanins, can be associated with receptors inducing internalization. 

Response 1: We thank the reviewer for the suggestions, and therefore, have added several highlighted sections to the discussion section to address the reviewer’s concern. We have added two paragraphs, lines 647-674, to include information about receptors and coreceptors, especially tetraspanins.

Comment 2: There are only two tables and no figures, which, for a review paper of this dimension, is not adequate. At least two figures should be included, including constructs and interactions of the constructs.

Response 2: We thank the reviewer for the suggestion and have added two figures that depict engineered viral tropism and their applications from our collaborative projects. The new authors are contributing researchers that played a significant role in the production of figures. Dr. Robert Petrovich, Molly Cook, and Erica Scappini constructed and purified the BacMam virus and assisted in performing the experiments and imaging for Figure 1. Dr. Kedar Sharma provided the Cryo-EM image in Figure 2. The information about Figure 1 was added in lines 64-69 of text. Lines 421-425 in text introduce Figure 2. 

Comment 3: Part of the text should also be checked since there are some similarities with previously published research.

Response 3: We respect the reviewer’s concern about similarities with previously published research. Therefore, we ran the revised manuscript through iThenticate software with a help of a NIH librarian and generated a similarity report that showed <1% similarity to published work. We do understand the reviewer’s concern. We had to often reiterate background information to create a backdrop for sharing more recent information. If there is a specific section of the text that you would like us to remove, we will definitely honor your wishes.       

Comment 4: Finally, what are the possible limitations? Is there any certainty of safe molecular modifications?

Response 4: This is a great and insightful suggestion. We thank the reviewer. We have added several highlighted sections to the discussion section in lines from 604-629 and 647-700 to address possible concerns and limitations of the engineered tropism and offer possible future directions in developing new strategies. Reviewer’s comment has definitely helped us improve the manuscript.  

Round 2

Reviewer 2 Report

Comments and Suggestions for Authors

The manuscript has been improved. I thank the authors for responding all my queries. The article is suitable for publication.